# Trajectory balance:
# Improved credit assignment in GFlowNets

**Nikolay Malkin**
Mila, Université de Montréal
Montréal, Québec, Canada

**Moksh Jain**
Mila, Université de Montréal
Montréal, Québec, Canada

**Emmanuel Bengio**
Mila, McGill University, Recursion
Montréal, Québec, Canada

**Chen Sun**
Mila, Université de Montréal
Montréal, Québec, Canada

**Yoshua Bengio**
Mila, Université de Montréal
Montréal, Québec, Canada
`{nikolay.malkin,moksh.jain,chen.sun,yoshua.bengio}@mila.quebec`
`emmanuel.bengio@recursionpharma.com`

## Abstract

Generative flow networks (GFlowNets) are a method for learning a stochastic policy for generating compositional objects, such as graphs or strings, from a given unnormalized density by sequences of actions, where many possible action sequences may lead to the same object. We find previously proposed learning objectives for GFlowNets, *flow matching* and *detailed balance*, which are analogous to temporal difference learning, to be prone to inefficient credit propagation across long action sequences. We thus propose a new learning objective for GFlowNets, *trajectory balance*, as a more efficient alternative to previously used objectives. We prove that any global minimizer of the trajectory balance objective can define a policy that samples exactly from the target distribution. In experiments on four distinct domains, we empirically demonstrate the benefits of the trajectory balance objective for GFlowNet convergence, diversity of generated samples, and robustness to long action sequences and large action spaces.

## 1 Introduction

Generative flow networks [GFlowNets; 3, 4] are models that exploit generalizable structure in an energy function $\mathcal{E}$ to amortize sampling from the corresponding probability density function on a space of compositional objects $\mathcal{X}$, for example, graphs composed of nodes and edges. A GFlowNet learns a stochastic policy that generates such structured objects by producing a stochastic sequence of *actions* that incrementally modify a partial object (*state*), e.g., by adding an edge or a node to a graph, starting from a universal initial state (like an empty graph). A special 'exit' action signals that the construction of the object $x \in \mathcal{X}$ is completed, and the policy is trained so as to make the likelihood of generating $x$ proportional to the given unnormalized probability or reward $R(x) = e^{-\mathcal{E}(x)}$.

Like other models in deep reinforcement learning [RL; 25], GFlowNets are trained with a parametric policy that can be given desired inductive biases (e.g., a particular deep net architecture) and allows generalization to states not seen in training. Natural domains for applying GFlowNets are those where exact sampling is intractable and local exploration (MCMC) methods perform poorly, but

36th Conference on Neural Information Processing Systems (NeurIPS 2022).

diversity of samples is desired [3, 32, 14, 8]. For example, GFlowNets have been used [3] to generate graphical descriptions of molecules by incremental addition of simple building blocks, where the reward $R(x)$ is the estimated strength of binding the constructed molecule to a protein target: the number of candidates grows rapidly with the molecule size and the reward has many separated modes. Like all RL models that iteratively sample action sequences for training, GFlowNets pose the learning challenges of exploration/exploitation and credit assignment, i.e., propagation of a reward signal over an action sequence [27, 2, 17]. *The efficiency of credit assignment and training in GFlowNets is the focus of the present paper.*

The learning problem solved by GFlowNets also has two fundamental differences with the standard reward-maximization paradigm of RL. First, a GFlowNet aims to make the likelihood of reaching a terminating state proportional to the reward, not to concentrate it at a maximal-reward state. Thus, a GFlowNet must model the diversity in the target distribution, not only its dominant mode. Reward maximization in RL can be turned into sampling proportionally to the reward with appropriate entropy maximization regularization, if there is only one way to reach every state [3]. The second difference with reward-maximization in RL is indeed that the GFlowNet training objectives still lead to correct sampling even when multiple action sequences lead to the same terminating state. Note that the likelihood of reaching a state is the sum of likelihoods of all action sequences leading to it, and that the number of such paths may be exponential in their length.

The set of all achievable sequences of actions and states can be conceptually organized in a directed graph $G = (\mathcal{S}, \mathcal{A})$ in which the vertices $\mathcal{S}$ are states (some of them designated as terminal states, in bijection with $\mathcal{X}$) and the edges $u{\to}v$ in $\mathcal{A}$ each correspond to applying an action while in a state $u \in \mathcal{S}$ and landing in state $v$. In [3], a GFlowNet is described by a nonnegative function on the edges, called the *edge flow* $F : \mathcal{A} \to \mathbb{R}_{\geq 0}$, where $F(u{\to}v)$ is an *unnormalized* likelihood of taking the action that modifies state $u$ to state $v$. The *GFlowNet policy* samples the transition $u{\to}v$ from state $u$ with probability $F(u{\to}v)/\sum_{v'} F(u{\to}v')$ where the denominator sums over the outgoing edges from $u$. By analogy with the classical notion of flows in networks [9], one can think of this flow like the amount of water flowing through an edge (like a pipe) or a state (like a tee, where pipes meet). It is shown that this GFlowNet policy samples $x$ proportionally to $R(x)$ if $F$ satisfies a set of linear *flow matching* constraints (a conservation law: the sum of flows into a state should equal the sum of flows out of it). These constraints are converted into a temporal difference-like objective that can be optimized with respect to the parameters of a neural net that approximates $F$. An alternative objective based on *detailed balance* constraints was proposed in [4]. These objectives, however, like temporal-difference learning, can suffer from slow credit assignment [27, 2, 17].

The **main contribution** of this work (§3) is a new parametrization and objective for GFlowNets. This objective, which we call *trajectory balance*, is computed on sampled full action sequences (trajectories) from the initial state to a terminal state, unlike the flow matching and detailed balance objectives. We prove that global minimization of trajectory balance implies that the learned action policy samples proportionally to $R$. We also empirically show that *the trajectory balance objective accelerates training convergence* relative to previously proposed objectives, improves the learned sampling policy with respect to metrics of diversity and divergence from the reward function, and allows learning GFlowNets that generate sequences far longer than was possible before. As a **secondary contribution**, we perform the first empirical validation of the detailed balance training objective. Comparative evaluation of the three GFlowNet objectives and non-GFlowNet baselines is performed on four domains illustrating different features of the reward landscape:

- **Hypergrid** (§5.1), an illustrative synthetic environment with modes separated by wide troughs;
- **Molecule synthesis** (§5.2), a practical graph generation problem, where the trajectory balance objective leads to significant computational speed-ups and more diverse generated candidates;
- **Sequence generation** (§5.3), where we show the robustness of trajectory balance to large action spaces and long action sequences on synthetic data and real AMP sequence data.

Since the initial appearance of this work on arXiv, several published papers and preprints have used trajectory balance and its generalizations successfully in various applications [32, 14, 8, 18].

## 2 Preliminaries

### 2.1 Markovian flows

We give some essential definitions, following §2 of [4]. Fix a directed acyclic graph $G = (\mathcal{S}, \mathcal{A})$ with state space $\mathcal{S}$ and action space $\mathcal{A}$. Let $s_0 \in \mathcal{S}$ be the special *initial (source) state*, the only state with no incoming edges, and designate vertices with no outgoing edges as *terminal* (sinks) [1]. We call the vertices *states*, the edges *actions*, the states reachable through outgoing edges from a state its *children*, and the sources of its incoming edges its *parents*.

A *complete trajectory* is a sequence of transitions $\tau = (s_0 \to s_1 \to \ldots \to s_n)$ going from the initial state $s_0$ to a terminal state $s_n$ with $(s_t \to s_{t+1}) \in \mathcal{A}$ for all $t$. Let $\mathcal{T}$ be the set of complete trajectories. A *trajectory flow* is a nonnegative function $F : \mathcal{T} \to \mathbb{R}_{\geq 0}$. With our water analogy, it could be the number of water molecules travelling along this path (the units don't matter because the flow function can be scaled arbitrarily, since we normalize them to get probabilities). For any state $s$, define the state flow $F(s) = \sum_{s \in \tau} F(\tau)$, and, for any edge $s \to s'$, the edge flow

$$F(s \to s') = \sum_{\tau = (\ldots \to s \to s' \to \ldots)} F(\tau). \tag{1}$$

As a consequence of this definition, the *flow matching* constraint (incoming flow = outgoing flow) is satisfied for all states $s$ that are not initial or terminal:

$$F(s) = \sum_{(s'' \to s) \in \mathcal{A}} F(s'' \to s) = \sum_{(s \to s') \in \mathcal{A}} F(s \to s'). \tag{2}$$

A nontrivial (i.e., not identically zero) trajectory flow $F$ determines a distribution $P$ over trajectories,

$$P(\tau) = \frac{1}{Z} F(\tau), \quad Z = F(s_0) = \sum_{\tau \in \mathcal{T}} F(\tau). \tag{3}$$

The trajectory flow $F$ is *Markovian* if there exist action distributions $P_F(-|s)$ over the children of each nonterminal state $s$ such that the distribution $P$ has a factorization

$$P(\tau = (s_0 \to \ldots \to s_n)) = \prod_{t=1}^{n} P_F(s_t | s_{t-1}). \tag{4}$$

Equivalently ([4], Prop. 3) there are distributions $P_B(-|s)$ over the parents of each noninitial state $s$ such that for any terminal $x$,

$$P(\tau = (s_0 \to \ldots \to s_n) | s_n = x) = \prod_{t=1}^{n} P_B(s_{t-1} | s_t). \tag{5}$$

If $F$ is a Markovian flow, then $P_F$ and $P_B$ can be computed in terms of state and edge flows:

$$P_F(s'|s) = \frac{F(s \to s')}{F(s)}, \quad P_B(s|s') = \frac{F(s \to s')}{F(s')}, \tag{6}$$

supposing denominators do not vanish. We call $P_F$ and $P_B$ the *forward policy* and *backward policy* corresponding to $F$, respectively. These relations are summarized by the *detailed balance* constraint

$$F(s) P_F(s'|s) = F(s') P_B(s|s'). \tag{7}$$

**Uniqueness properties.** A Markovian flow is uniquely determined by an edge flow, i.e., a nontrivial choice of nonnegative value on every edge satisfying the flow matching constraint (2). By Corollary 1 of [4], a Markovian flow is also uniquely determined by either of

- a constant $Z = F(s_0) > 0$ and a distribution $P_F(-|s)$ over children of every nonterminal state; or
- a nontrivial choice of nonnegative state flows $F(x)$ for every terminal state $x$ and a choice of distribution $P_B(-|s)$ over parents of every noninitial state.

---

[1] [3] allowed terminal states with outgoing edges. The difference is easily overcome by augmenting every such state $x$ by a new terminal state $x^\top$ with a stop action $x \to x^\top$.

## 2.2 GFlowNets

Suppose that a nontrivial nonnegative reward function $R : \mathcal{X} \to \mathbb{R}_{\geq 0}$ is given on the set of terminal states. GFlowNets [3] aim to approximate a Markovian flow $F$ on $\overline{G}$ such that

$$F(x) = R(x) \quad \forall x \in \mathcal{X}. \tag{8}$$

We adopt the broad definition that a GFlowNet is any learning algorithm consisting of:

- a model capable of providing the initial state flow $Z = F(s_0)$ as well as the forward action distributions $P_F(-|s)$ for any nonterminal state $s$ (and therefore, by the above, uniquely but possibly in an implicit way determining a Markovian flow $F$);
- an objective function, such that if the model is capable of expressing any action distribution and the objective function is globally minimized, then the constraint (8) is satisfied for the corresponding Markovian flow $F$.

The forward policy of a GFlowNet can be used to sample trajectories from the corresponding Markovian flow $F$ by iteratively taking actions according to policy $P_F(-|s)$. If the objective function is globally minimized, then the likelihood of terminating at $x$ is proportional to $R(x)$.

In general, an objective optimizing for (8) cannot be minimized directly because $F(x)$ is a sum over all trajectories leading to $x$, and computing it may not be practical. Therefore, two local objectives – *flow matching* and *detailed balance* – have previously been proposed.

**Flow matching objective [3].** A model $F_\theta(s, s')$ [2] with learnable parameters $\theta$ approximates the edge flows $F(s \to s')$. The corresponding forward policy is given by $P_F(s'|s; \theta) \propto F_\theta(s, s')$ (Eq. (6)). Denote the corresponding Markovian flow by $F_\theta$ and distribution over trajectories by $P_\theta$. The parameters are trained to minimize the error in the flow matching constraint (2) for all noninitial and nonterminal nodes $s$:

$$\mathcal{L}_{\text{FM}}(s) = \left( \log \frac{\sum_{(s'' \to s) \in \mathcal{A}} F_\theta(s'', s)}{\sum_{(s \to s') \in \mathcal{A}} F_\theta(s, s')} \right)^2 \tag{9}$$

and a similar objective $\mathcal{L}'_{\text{FM}}$ pushing the inflow at $x \in \mathcal{X}$ to equal $R(x)$ at terminal nodes $x$. This objective is optimized for nonterminal states $s$ and terminal states $x$ from trajectories sampled from a training policy $\pi_\theta$. Usually, $\pi_\theta$ is chosen to be a tempered (higher temperature) version of $P_F(-|s, \theta)$, which also helps exploration during training. The parameters are updated with stochastic gradient

$$\mathbb{E}_{\tau = (s_0 \to \dots \to s_n) \sim \pi_\theta} \nabla_\theta \left[ \sum_{t=1}^{n-1} \mathcal{L}_{\text{FM}}(s_t) + \mathcal{L}'_{\text{FM}}(s_n) \right]. \tag{10}$$

As per Proposition 10 of [4], if the training policy $\pi_\theta$ has full support, and a global minimum of the expected loss (9) over states on trajectories sampled from $\pi_\theta$ is reached, then the GFlowNet samples from the target distribution (i.e., $F_\theta$ satisfies (8)).

**Detailed balance objective [4].** A neural network model with parameters $\theta$ has input $s$ and three kinds of outputs: an estimated state flow $F_\theta(s)$, an estimated distribution over children $P_F(-|s; \theta)$, and an estimated distribution over parents $P_B(-|s; \theta)$. The policy $P_F(-|-; \theta)$ and the initial state flow $F_\theta(s_0)$ uniquely determine a Markovian flow $F_\theta$, which is not necessarily compatible with the estimated backward policy $P_B(-|-; \theta)$. The error in the detailed balance constraint (7) is optimized on actions $(s \to s')$ between nonterminal nodes seen along trajectories sampled from the training policy:

$$\mathcal{L}_{\text{DB}}(s, s') = \left( \log \frac{F_\theta(s) P_F(s'|s; \theta)}{F_\theta(s') P_B(s|s'; \theta)} \right)^2, \tag{11}$$

and a similar constraint $\mathcal{L}'_{\text{DB}}(s, x)$ is optimized at actions leading to terminal nodes. Similarly to flow matching, the parameters are updated with stochastic gradient

$$\mathbb{E}_{(s_0 \to \dots \to s_n) \sim \pi_\theta} \nabla_\theta \left[ \sum_{t=1}^{n-1} \mathcal{L}_{\text{DB}}(s_{t-1}, s_t) + \mathcal{L}'_{\text{DB}}(s_{n-1}, s_n) \right] \tag{12}$$

along trajectories sampled from a training policy $\pi_\theta$. By Proposition 6 of [4], a global minimum of the expected detailed balance loss under a $\pi_\theta$ with full support specifies a GFlowNet that samples from the target distribution, i.e., the flow $F_\theta$ satisfies (8).

---

[2] In practice, it is convenient and more economical to provide a representation of $s$ to the neural net, which simultaneously outputs the flows $F_\theta(s, s')$ for all $s'$ that are reachable by an action from $s$.

---

**Algorithm 1** Training a GFlowNet with trajectory balance

---
**input** Reward function $R : \mathcal{X} \to \mathbb{R}_{>0}$, model and optimizer hyperparameters
 1: Initialize models $P_F, P_B, Z$ with parameters $\theta$
 2: **repeat**
 3:    Sample trajectory $\tau = (s_0 \to \cdots \to s_n)$ from policy $P_F(-|-; \theta)$ or a tempered version of it
 4:    $\theta \leftarrow \theta - \eta \nabla_\theta \mathcal{L}_{\text{TB}}(\tau)$ {gradient update on (14)}
 5: **until** convergence monitoring on running $\mathcal{L}_{\text{TB}}(\tau)$

---

**Remarks.** In some problems, such as autoregressive sequence generation (§5.3), the directed graph $G$ is a tree, so each state has only one parent. In this case, $P_B$ is trivial and the detailed balance objective reduces to the flow matching objective, which in turn can be shown to be equivalent to Soft Q-Learning [12, 5] with temperature $\alpha = 1$, a uniform $q_{\mathbf{a}'}$, and $\gamma = 1$.

## 3  Trajectory balance

Let $F$ be a Markovian flow and $P$ the corresponding distribution over complete trajectories, defined by (3), and let $P_F$ and $P_B$ be forward and backward policies determined by $F$. A direct algebraic manipulation of Eqs. (3,4,5) gives the *trajectory balance constraint* for any complete trajectory $\tau = (s_0 \to s_1 \to \ldots \to s_n = x)$:

$$Z \prod_{t=1}^n P_F(s_t|s_{t-1}) = F(x) \prod_{t=1}^n P_B(s_{t-1}|s_t), \tag{13}$$

where we have used that $P(s_n = x) = \frac{F(x)}{Z}$.

As explained in §A.2, the trajectory balance constraint (13) and the detailed balance constraint (7) are special cases of one general constraint, which has been studied as a training objective in [18].

**Trajectory balance as an objective.** We propose to convert (13) into an objective to be optimized along trajectories sampled from a training policy. Suppose that a model with parameters $\theta$ outputs estimated forward policy $P_F(-|s; \theta)$ and backward policy $P_B(-|s; \theta)$ for states $s$ (just as for detailed balance above), as well as a global scalar $Z_\theta$ estimating $F(s_0)$. The scalar $Z_\theta$ and forward policy $P_F(-|-; \theta)$ uniquely determine an implicit Markovian flow $F_\theta$.

For a trajectory $\tau = (s_0 \to s_1 \to \ldots \to s_n = x)$, define the *trajectory loss*

$$\mathcal{L}_{\text{TB}}(\tau) = \left( \log \frac{Z_\theta \prod_{t=1}^n P_F(s_t|s_{t-1}; \theta)}{R(x) \prod_{t=1}^n P_B(s_{t-1}|s_t; \theta)} \right)^2. \tag{14}$$

If $\pi_\theta$ is a training policy – usually that given by $P_F(-|-; \theta)$ or a tempered version of it – then the trajectory loss is updated along trajectories sampled from $\pi_\theta$, i.e., with stochastic gradient

$$\mathbb{E}_{\tau \sim \pi_\theta} \nabla_\theta \mathcal{L}_{\text{TB}}(\tau). \tag{15}$$

The full algorithm, with batch size of 1, is presented as Algorithm 1 and its correctness is guaranteed by the following.

**Proposition 1.** *Let $R$ be a positive reward function on $\mathcal{X}$.*

 (a) *If $P_F(-|-; \theta)$, $P_B(-|-; \theta)$, and $Z_\theta$ are the forward and backward policies and normalizing constant of a Markovian flow $F$ satisfying (8), then $\mathcal{L}_{\text{TB}}(\tau) = 0$ for all complete trajectories $\tau$.*
 (b) *Conversely, suppose that $\mathcal{L}_{\text{TB}}(\tau) = 0$ for all complete trajectories $\tau$. Then the corresponding Markovian flow $F_\theta$ satisfies (8), and $P_F(-|-; \theta)$ samples proportionally to the reward.*

The proof is given in §A.1. In particular, if $\pi_\theta$ has full support and $\mathbb{E}_{\tau \sim \pi_\theta} \mathcal{L}_{\text{TB}}(\tau)$ is globally minimized over all forward and backward policies $(P_F, P_B)$ and normalizing constants $Z$, then the corresponding Markovian flow $F_\theta$ satisfies (8) and $P_F(-|-; \theta)$ samples proportionally to the reward. (The positivity assumption on $R$ is necessary to avoid division by 0 in (14), but can be relaxed by introduction of smoothing constants, just as was done for the losses proposed in [3, 4].)

**Remarks. (1)** As discussed in §2, in the case of auto-regressive generation, $G$ is a directed tree, where each $s \in \mathcal{S}$ has a single parent state. In this case $P_B$ is trivially $P_B = 1, \forall s \in \mathcal{S}$. We get

$$\mathcal{L}_{\mathrm{TB}}(\tau) = \left( \log \frac{Z_\theta \prod_{t=1}^{n} P_F(s_t|s_{t-1};\theta)}{R(x)} \right)^2 \tag{16}$$

**(2)** We found it beneficial to parametrize $Z$ in the logarithmic domain ($\log Z$ is the trainable parameter) and output logits for $P_F(-|s;\theta)$ and $P_B(-|s;\theta)$, so that all products in (14) become sums under the logarithm. This is consistent with the log-domain parametrization of flows in [3]. In addition, we found it helpful to set a higher learning rate for $Z$ than for the parameters of $P_F$ and $P_B$.[3]

### 3.1 Canonical choice of reward-matching flow

The constraint (8), in general, does not have a unique solution: if the underlying undirected graph of $G$ has cycles, there may be multiple Markovian flows whose corresponding action policies sample proportionally to the reward. However, by the uniqueness properties, for any choice of backward policy $P_B(-|-)$, there is a unique flow satisfying (8), and thus a unique corresponding forward policy $P_F(-|-)$ for states with nonzero flow. (See Fig. 1.)

In some settings, it may be beneficial to *fix* the backward policy $P_B$ and train only the parameters giving $P_F$ and $Z_\theta$. For example, it may difficult to construct a model that outputs a distribution over the parents of a given input state (e.g., for the molecule domain (§5.2), it is hard to force invariance to molecule isomorphism). A natural choice is to set $P_B(-|s)$ to be uniform over all the parents of a state $s$, i.e., $P_B(-|s) = 1/\#\{s' \mid (s' \rightarrow s) \in \mathcal{A}\}$.

## 4 Related work

**Reinforcement learning.** GFlowNets are trained to sample proportionally the reward rather than maximize it as usual in RL. However, on tree-structured DAGs (autoregressive generation) are equivalent to RL with appropriate entropy regularization or soft Q-learning and control as inference [5, 12, 13]. The experiments and discussion of [3] show how these methods can fail badly in the general DAG case well handled by GFlowNets. Signal propagation over sequences of several actions in trajectory balance is also related to losses used in RL computed on subtrajectories [22].

**Local exploration vs. amortized generalization to unseen modes.** GFlowNets are also related to MCMC methods for sampling from unnormalized densities. While there has been work on accelerating or partially amortizing sampling from unnormalized densities over discrete spaces when exact sampling is intractable [11, 7], some of it domain- or problem-specific [30], GFlowNets treat the compositional structure in data as a learning problem (enabling generalization to unseen modes), not as a bias to build in to the sampler. Thus, the cost is amortized and borne by the learning of that structure through sampling, not through search at generation time.

**Variational inference.** GFlowNets are connected with variational methods for fitting hierarchical generative models. The squared log-ratio loss proposed in [20] as a control variate in the optimization of an evidence lower bound can be seen as a special case of trajectory balance. See §A.3 for further discussion, in which we prove that on-policy optimization of trajectory balance is equivalent to minimization of a certain KL divergence. Two recent papers [31, 19] extend our observations.

## 5 Experiments

We evaluate the proposed trajectory balance objective against prior objectives for training GFlowNets as well as standard methods for learning policies that approximately sample objects proportionally to their rewards, like MCMC as well as against other RL techniques. Our experiments include the hypergrid and molecule synthesis tasks from [3] and two new tasks in which $G$ is a directed tree.

---

[3]Because the loss (14) is quadratic in $\log Z$, gradient updates on $\log Z$ are equivalent to setting it to a weighted moving average of the discrepancy between $\log \prod P_F$ and $\log(R(x) \prod P_B)$. Optimizers enhanced with momentum complicate things: we leave empirical investigation of these questions to future work.

## 5.1 Hypergrid environment

In this subsection, we study a synthetic hypergrid environment introduced by [3]. This task is easier than others we study, but we include it for completeness, and because it allows us to illustrate some interesting behaviours.[4]

In this environment, the nonterminal states $\mathcal{S}^\circ$ form a $D$-dimensional hypergrid with side length $H$:

$$\mathcal{S}^\circ = \{(s^1, \ldots, s^D) \mid s^d \in \{0, 1, \ldots, H-1\}, d = 1, \ldots, D\},$$

and actions are operations of incrementing one coordinate in a state by 1 without exiting the grid. The initial state is $(0, \ldots, 0)$. For every nonterminal state $s$, there is also a termination action that transitions to a corresponding terminal state $s^\top$ (cf. footnote 1). The reward at a terminal state $s^\top = (s^1, \ldots, s^d)^\top$ is given by

$$R(s^\top) = R_0 + 0.5 \prod_{d=1}^{D} \mathbb{I}\left[\left|\frac{s^d}{H-1} - 0.5\right| \in (0.25, 0.5]\right] + 2 \prod_{d=1}^{D} \mathbb{I}\left[\left|\frac{s^d}{H-1} - 0.5\right| \in (0.3, 0.4)\right]$$

where $\mathbb{I}$ is an indicator function and $R_0$ is a constant controlling the difficulty of exploration. This reward has peaks of height $2.5 + R_0$ near the corners of the hypergrid, surrounded by plateaux of height $0.5 + R_0$. These plateaux are separated by wide troughs with reward $R_0$. An illustration with $H = 8$ and $D = 2$ is shown in the left panel of Fig. 1. This environment evaluates the ability of a GFlowNet to generalize from visited states to infer the existence of yet-unvisited modes.

We train GFlowNets with the detailed balance (DB) and trajectory balance (TB) objectives with different $H$, $D$, and $R_0$, in addition to reproducing the flow matching (FM) experiments and non-GFlowNet baselines based upon [3]'s published code. Our GFlowNet policy model is a multilayer perceptron (MLP) that accepts as input a one-hot encoding of a state $s$ (with the goal of enabling generalization) and outputs logits of the forward and backward policies $P_F(-|-; \theta)$ and $P_B(-|-; \theta)$ (as well as the estimated state flow $F_\theta(s)$ in the case of DB). The forward policy, backward policy, and state flow models share all but the last weight matrix of the MLP. This is consistent with [3]'s model, where an identical architecture accepted $s$ as input and output estimated flows $F_\theta(s, s')$ for all children $s'$ of $s$. Details are given in §B.1.

We consider a 4-dimensional grid with $H = 8$ and and a 2-dimensional grid with $H = 64$. The two grids have the same number of terminal states, but the 2-dimensional grid has longer expected trajectory lengths. For both grid sizes, we consider $R_0 = 0.1, 0.01, 0.001$, with smaller $R_0$ giving environments that are more difficult to explore due to the lower likelihood for models to cross the low-reward valley. For the models trained with DB and TB, we also explore the effect of fixing the backward policy to be uniform (§3.1).

**Results.** In Fig. 2, we plot the evolution over the course of training of the $L_1$ error between the true reward distribution (the reward $R(x)$ normalized over all possible terminal states $x \in \mathcal{X}$) and the empirical distribution of the last $2 \cdot 10^5$ visited states for all settings (which would have a probability of 0 on $x$'s not visited). Although convergence to the same stable minimum is achieved by all models and settings, DB and TB training tend to converge faster than FM, with a slight benefit of TB over detailed balance in the 4-D environment.

**Effect of uniform $P_B$.** Note the difference in learning speed between models with fixed uniform backward policy $P_B$ and models with learned $P_B$. As noted in §3.1, when $P_B$ is fixed, there is a unique $P_F(-|-; \theta)$ that globally minimizes the objective, and it may be approached slowly. However, if $P_B$ and $P_F$ are permitted to evolve jointly, they may more quickly approach one of the many optimal solutions. This is confirmed by the much faster convergence of models with learned $P_B$ on the $64 \times 64$ grid. We have observed that, especially for large grid sizes, when $P_B$ and $P_F$ are both learned, the model has a bias towards first taking all actions in one coordinate direction, then proceeding in the other direction until terminating (as in the right panel of Fig. 1), perhaps because a constant distribution over two actions ('continue to the right' and 'terminate') can be modeled with higher precision over a large portion of the grid than the complex position-dependent distribution as shown in the centre panel of Fig. 1.

---

[4]Code: https://gist.github.com/malkin1729/9a87ce4f19acdc2c24225782a8b81c15.

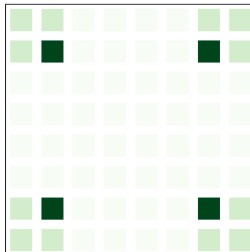 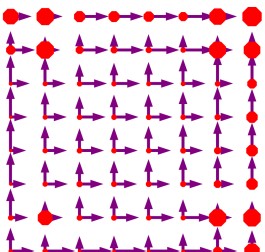 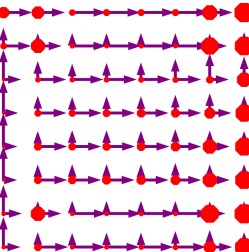

Figure 1: *Left:* The reward function on an $8 \times 8$ grid environment (§5.1) with $R_0 = 0.1$. *Centre and right:* Two forward action policies – with fixed uniform $P_B$ and with a learned non-uniform $P_B$ – that sample from this reward. The lengths of arrows pointing up and right from each state are proportional to the likelihoods of the corresponding actions under $P_F$, and the sizes of the red octagons are proportional to the termination action likelihoods.

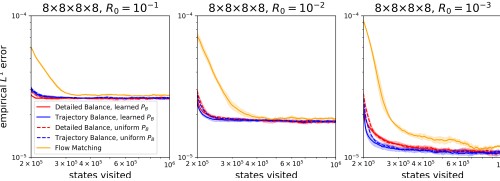 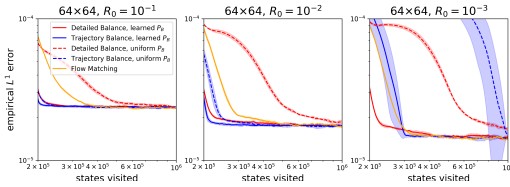

Figure 2: Empirical $L^1$ error between true and sampled state distributions on the grid environment with varying grid size and $R_0$. Mean and standard error over 5 seeds. The curves for PPO and MCMC baseline would lie outside the plot bounds.

## 5.2 Small drug molecule synthesis

Next, we consider the molecule generation task [30, 15, 16, 10, 24] introduced for GFlowNets in [3]. We extend [3]'s published code with an implementation of the TB and DB objectives.[5]

The goal is to generate molecules, in the form of graphs, with a low binding energy to the 4JNC inhibitor of the sEH (soluble epoxide hydrolase) protein. The graphs generated are junction trees [15] of a vocabulary of building blocks. The reward is defined as the normalized negative binding energy as predicted by a *proxy* model, itself trained to predict energies computed via docking simulations [26]. The maximum trajectory length is 8, with the number of actions varying between around 100 and 2000 (the larger a molecule, the more possible additions exist), making $|\mathcal{X}|$ about $10^{16}$.

**Results.** We plot in Fig. 3 (left and centre) the correlation of log-reward and log-sampling probability (the likelihood that a trajectory sampled from the learned policy terminates at $x$) for GFlowNets trained using TB, FM and DB. This correlation is significantly higher for models trained with TB. The points $x$ shown are from a fixed held-out set to which the GFlowNets do not have access in training. Note that a perfect model would have correlation 1, as $\log R(x)$ and $\log p_\theta(x)$ would differ by a constant (equal to $\log Z$) that is independent of $x$.

In Fig. 3 (right) we plot the average pairwise Tanimoto similarity [1] for the 1000 highest-reward samples generated over the course of training. We see that TB consistently generates more diverse molecules than FM. These results showcase the benefits of TB, not only for faster temporal credit assignment, but for generalization and diversity. In addition, TB has up to $5\times$ runtime speedup over FM as the enumeration of parents is not needed. See §B.2 for further discussion.

## 5.3 Autoregressive sequence generation

Finally, we evaluate the TB objective on the task of autoregressive sequence generation. In §5.3.1, we study the effect of trajectory length and action space size on the learning dynamics in GFlowNets. In §5.3.2, we consider the more realistic task of generating peptides (short protein sequences with anti-microbial properties) and evaluate GFlowNets against standard RL and MCMC baselines.

---

[5]Code: https://github.com/GFNOrg/gflownet/tree/trajectory_balance.

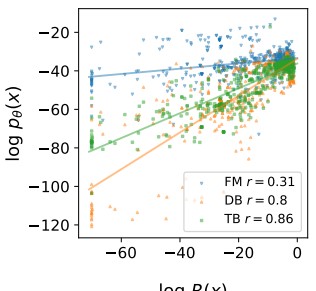 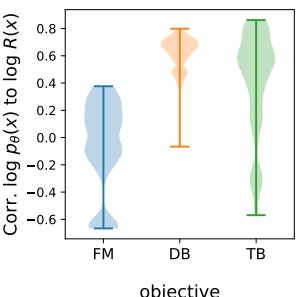 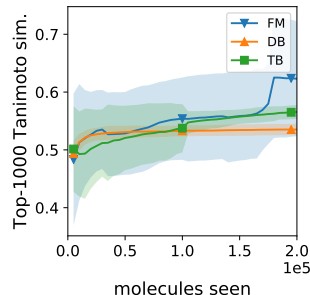

Figure 3: *Left, Centre:* Pearson correlations between rewards and sampling probability. $\log p_\theta(x)$ is the log-likelihood that a trajectory sampled from the learned policy $P_F(-|-;\theta)$ terminates at $x$. *Left:* Scatter plot on a test set of $x$'s for the best hyperparameters of TB, FM, and DB. *Centre:* Violin plot of correlations for 16 hyperparameter settings and 3 seeds for each setting, showing TB being capable of fitting better. *Right:* Average pairwise Tanimoto similarity for the top 1000 samples generated by GFlowNets as training progresses. Lines are the average across runs, shaded regions the standard deviation. Models trained with TB have consistently lower similarity than those with FM, hence greater diversity. We hypothesize that the higher variance, in correlation and diversity, of TB relative to DB is related to high variance of the stochastic gradient; see [18].

### 5.3.1 Bit sequences

**Task.** The goal is to generate bit sequences of a fixed length $n = 120$ ($\mathcal{X} = \{0,1\}^n$), where the reward is designed to have modes at a given fixed set $M \subset \mathcal{X}$ that is unknown to the learner. The reward for a sequence $x$ is defined as $R(x) = \exp(-\min_{y \in M} d(x,y))$, where $d$ is the edit distance. We describe the procedure to generate $M$ in §B.3.

For different integers $k$ dividing $n$, we design action spaces for left-to-right generation of sequences in $\mathcal{X}$, where a complete trajectory has $\frac{n}{k}$ actions and each action appends a $k$-bit 'word' to the end of a partial sequence. A forward policy on this state space is the same an autoregressive sequence model over a vocabulary of size $2^k$. Varying $k$ while fixing $\mathcal{X}$ and $M$ allows us to study the effect of the tradeoff between trajectory lengths ($\frac{n}{k}$) and the action space sizes ($|V| = 2^k$) without changing the underlying probabilistic modeling problem.

We compare GFlowNets trained with the TB objective against GFlowNets trained with the FM objective (equivalent to DB and Soft Q-Learning in this case) and two non-GFlowNet baselines: A2C with Entropy Regularization [29, 21], Soft Actor-Critic [13, 6] and MARS [30]. We use a Transformer-based architecture [28] across all the methods. See §B.3 for details.

To evaluate the methods we use (1) Spearman correlation between the probability of generating the sequence $p(x) = F(x)/Z$ and its reward $R(x)$ on a test set sampled approximately uniformly over the possible values of the reward, (2) number of modes captured (number of reference sequences from $M$ for which a candidate within a distance $\delta$ has been generated). In our experiments, $n = 120$, $|M| = 60$, $k \in \{1, 2, 4, 6, 8, 10\}$, and $\delta = 20$.

**Results.** Fig. 4 (left) presents the results for the Spearman correlation evaluation. We observe that GFlowNets trained with the TB objective learn policies that correlate best with the reward $R(x)$ across all action spaces. In particular, we observe the effect of inefficient credit assignment in GFlowNets trained with FM, as the correlation improves with increasing $k$, i.e., shorter trajectories. On the other hand, large action spaces also hurt GFlowNets trained with the FM objective, while the TB objective is robust to them. Additionally, we can observe in Fig. 4 (right) that for fixed $k$, GFlowNets trained with TB discover more modes faster than other methods.

### 5.3.2 Anti-Microbial Peptide (AMP) generation

In this section, we consider the practical task of generating peptide sequences that have anti-microbial activity. The goal is to generate a protein sequence (where the vocabulary consists of 20 amino acids and

Table 1: Results on the AMP generation task.

|  | Top 100 Reward | Top 100 Diversity |
|---|---|---|
| GFN-$\mathcal{L}_{TB}$ | **0.85** $\pm 0.03$ | **18.35** $\pm 1.65$ |
| GFN-$\mathcal{L}_{FM}$/$\mathcal{L}_{DB}$ | $0.78 \pm 0.05$ | $12.61 \pm 1.32$ |
| SAC | $0.80 \pm 0.01$ | $8.36 \pm 1.44$ |
| AAC-ER | $0.79 \pm 0.02$ | $7.32 \pm 0.76$ |
| MCMC | $0.75 \pm 0.02$ | $12.56 \pm 1.45$ |

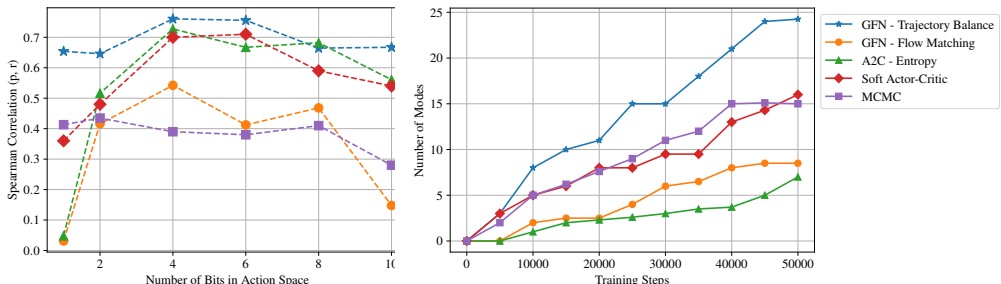

Figure 4: *Left:* Spearman correlation of the sampling probability under different learned policies and reward on a test set, plotted against the number of bits $k$ in the symbols in $V$ in the bit sequence generation task. GFlowNets trained with trajectory balance learn policies that have the highest correlation with the reward $R(x)$ and are robust to length and vocabulary size. *Right:* Number of modes discovered over the course of training on the bit sequence generation task with $k = 1$. GFlowNets trained with trajectory balance discover more modes faster.

a special end-of-sequence action), with maximum length $60$. We take $6438$ known AMP sequences and $9522$ non-AMP sequences from the DBAASP database [23]. We then train a classifier on this dataset, using $20\%$ of the data as a validation set. The probability output by this model for a sequence to be classified as an AMP is used as the reward $R(x)$ in our experiments.

The state and actions are designed just as in the previous experiment, with each action appending a symbol to the right of a state. We again compare TB and FM, as well as A2C with entropy regularization, SAC and MCMC as baselines. We again use Transformers for all the experiments on this task; see further details in Appendix B.4. We generate $2048$ sequences from each method, and pick the top $100$ sequences ranked by their reward $R(x)$. As metrics, we use the mean reward for these $100$ sequences and the average pairwise edit distance among them as a measure of *diversity*.

**Results.** We present the results in Table 1, where we observe that GFlowNets trained with TB outperform all baselines on both performance and diversity metrics.

## 6    Discussion and conclusion

We introduced a novel training loss for GFlowNets, trajectory balance (TB), which yields faster and better training than the previously proposed flow matching (FM) and detailed balance (DB) losses. We proved that this objective, when minimized, yields the desired GFlowNet property of sampling from the target distribution specified by an unnormalized reward function. This new loss was motivated by the observation that the FM and DB losses are local in the action sequence and may require many iterations for credit assignment to propagate to early actions: if a gradient update introduces a flow inconsistency at some state far from the initial state (such as when a novel high-reward state is sampled), the parent of this state must be visited before the update is propagated closer to the root, akin to the slow propagation of reward signals in temporal difference learning.

We empirically found that TB discovered more modes of the energy function faster and was more robust than FM and DB to the exponential growth of the state space, due in part to the lengths of sequences and in part to the size of the action space. A factor to consider when interpreting our experimental results is that because we use a neural net rather than a tabular representation of policies, the early states' transitions are informed by downstream credit assignment via parameter sharing. Early states also get many more visits because there are more possible states near the ends of sequences than near the initial state. Finally, TB trades off the advantage of immediately providing credit to early states with the disadvantage of relying on sampling of long trajectories and thus a potentially higher variance of the stochastic gradient. The high gradient variance is a possible limitation of TB in difficult environments, and ways to overcome it by interpolating between local and trajectory-level objectives have been studied in subsequent work [18].

All in all, we found that trajectory balance is a superior training objective in a broad set of experiments, making it the default choice for future work on GFlowNets.

## Acknowledgments

This research was enabled in part by computational resources provided by Compute Canada. All authors are funded by their primary academic institution. We also acknowledge funding from CIFAR, Samsung, IBM, Microsoft, and the Banting Postdoctoral Fellowship.

The authors are grateful to all the members of the Mila GFlowNet group, in particular to Dinghuai Zhang, for many fruitful research discussions, as well as to Yiheng Zhu for feedback on the published code. We also thank the anonymous reviewers for their comments.

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
