# A  Theoretical appendix

## A.1  Proof of Proposition 1

Recall Proposition 1:

**Proposition.** *Let $R$ be a positive reward function on $\mathcal{X}$.*

*(a) If $P_F(-|-;\theta)$, $P_B(-|-;\theta)$, and $Z_\theta$ are the forward and backward policies and normalizing constant of a Markovian flow $F$ satisfying (8), then $\mathcal{L}_{\mathrm{TB}}(\tau) = 0$ for all complete trajectories $\tau$.*

*(b) Conversely, suppose that $\mathcal{L}_{\mathrm{TB}}(\tau) = 0$ for all complete trajectories $\tau$. Then the corresponding Markovian flow $F_\theta$ satisfies (8), and $P_F(-|-;\theta)$ samples proportionally to the reward.*

*Proof.* Part (a) is an elementary manipulation of the trajectory balance constraint (13), with $R(x)$ substituted for $F(x)$ by the reward matching assumption (8).

Conversely, if $\mathcal{L}_{\mathrm{TB}}(\tau) = 0$ for all complete trajectories $\tau = (s_0 \to \to \cdots \to s_n = x)$, then the policies $P_F(-|-;\theta)$ and $P_B(-|-;\theta)$ satisfy the constraint

$$Z \prod_{t=1}^{n} P_F(s_t|s_{t-1};\theta) = R(x) \prod_{t=1}^{n} P_B(s_{t-1}|s_t;\theta). \tag{17}$$

Let $x$ be a terminal state. By iterating the law of total probability, we have

$$\sum_{\tau=(s_0 \to s_1 \to \ldots \to s_n = x)} \prod_{t=1}^{n} P_B(s_{t-1}|s_t;\theta) = 1. \tag{18}$$

(Each term in this sum is the conditional likelihood of $\tau$ conditioned on terminating at $s_n = x$ under the the Markovian flow $F_\theta'$ uniquely determined by setting terminal state flows $F_\theta'(x) = R(x)$ and backward policy $P_B(-|-;\theta)$, cf. the uniqueness properties.)

On the other hand, we have

$$F_\theta(x) = \sum_{\tau=(s_0 \to \ldots \to s_n)=x} F_\theta(\tau) \qquad \text{(by definition of state flows)}$$

$$= \sum_{\tau=(x_0 \to \ldots \to s_n)=x} Z \prod_{t=1}^{n} P_F(s_t|s_{t-1};\theta) \qquad \text{(by (4))}$$

$$= \sum_{\tau=(x_0 \to \ldots \to s_n)=x} R(x) \prod_{t=1}^{n} P_B(s_{t-1}|s_t;\theta) \qquad \text{(by (17))}$$

$$= R(x) \qquad \text{(by (18))}.$$

We conclude that $F_\theta$ satisfies (8), as desired.

(We remark that one can show in a similar way that $F_\theta(s \to s') = F_\theta'(s \to s')$ for all actions $(s \to s') \in \mathcal{A}$, and thus, by the uniqueness properties, $F_\theta = F_\theta'$, i.e., the forward and backward policies determine the same Markovian flow.)  $\square$

## A.2  Generalizations

The trajectory balance constraint (13) can be generalized to partial (not complete) trajectories, i.e., those that do not start at $s_0$ and end in a terminal state. Generalizations such as those we present here could be useful for a future goal of modularized or hierarchical GFlowNets, where each module (or low-level GFlowNet) can apply them to just the subsequences they have access to (cf. §9.4 and §10.2 in [4]).

**Subtrajectory balance.** If $\tau = (s_m \to s_{m+1} \to \ldots \to s_n)$ is a partial trajectory (i.e., $(s_t \to s_{t+1}) \in \mathcal{A}$ for all $t$), then, for any Markovian flow $F$ with forward and backward policies $P_F$ and $P_B$,

$$F(s_m) \prod_{t=m}^{n-1} P_F(s_{t+1} \mid s_t) = F(s_n) \prod_{t=m}^{n-1} P_B(s_t \mid s_{t+1}). \tag{19}$$

This can be derived by showing that both sides are equal to

$$\sum_{\tau=(\dots\to s_m\to s_{m+1}\to\dots\to s_n\to\dots)\in\mathcal{T}} F(\tau). \tag{20}$$

The trajectory balance constraint (13) is the special case of this for full trajectories, while the detailed balance constraint (7) is the special case of trajectories wth only one edge. This subtrajectory balance constraint can be converted into a learning objective: a model can output estimated state flows $F_\theta(s)$ only for certain nonterminal states $s$ ("hubs"), and the error in (19) optimized along segments of trajectories between these hubs. Thus the detailed balance loss corresponds to all nodes being hubs, and the trajectory balance loss corresponds to only the initial state $s_0$ being a hub.

Subtrajectory balance has been explored and was shown to have convergence benefits in a work that appeared while this paper was under review [18].

**Non-forward trajectories.** Trajectory balance has a more general form for trajectories that have a mix of forward and backward steps. Here we describe just one example: terminal-terminal paths that take several backward steps, then take several forward steps.

Let $s_1 = s_1'$ be any state (not necessarily a child of the GFlowNet's initial state $s_0$) and $(s_1\to s_2\to\dots\to s_n)$ and $(s_1'\to s_2'\to\dots\to s_{n'}')$ two trajectories from $s_0$ to terminal states. Then the following must hold for any Markovian flow $F$:

$$R(s_{n'}')\prod_{t=1}^{n'-1} P_B(s_t'\mid s_{t+1}')\prod_{t=1}^{n-1} P_F(s_{t+1}\mid s_t) = R(s_n)\prod_{t=1}^{n-1} P_B(s_t\mid s_{t+1})\prod_{t=1}^{n'-1} P_F(s_{t+1}'\mid s_t'). \tag{21}$$

That is, the path that goes "backward, then forward" from $s_n$ to $s_{n'}'$ must have the same likelihood no matter in which direction it is traversed, up to the ratio of rewards at the endpoints. A simple way to derive (21) is by writing the trajectory balance constraint for two paths from the GFlowNet's initial state to $s_n$ and $s_{n'}'$ that are identical until $s_1$ and then diverge, then dividing one constraint by the other. Notice that the flow $F(s_1)$ is not present here. Thus, (21) can be converted into an learning objective does not require a model to output any state flows (even the initial state flow $Z$).

Such terminal-terminal paths could also be used for exploration of $\mathcal{X}$ with MCMC-like local search algorithms [32]. The special case of 'one step back, two steps forward' paths was used for a graph generation problem in Bayesian structure learning [8].

### A.3  GFlowNets and variational methods

We build a connection between the TB loss for GFlowNets and a naïve variational approach to fitting sequential samplers.

Suppose that the backward policy $P_B$ is fixed, and suppose for ease of the derivation that it is known that $\sum_{x\in\mathcal{X}} R(x) = 1$. As in the main text, we write $P_B(\tau|x)$ for the likelihood of obtaining the reverse of the trajectory tau when sampling from the backward policy starting from $x$.

The on-policy trajectory balance objective has gradient with respect to the parameters of $P_F$:

$$\mathbb{E}_{(\tau,x)\sim P_F}\left[\nabla_\theta\left(\log\frac{R(x)P_B(\tau|x)}{P_F(\tau;\theta)}\right)^2\right] = \mathbb{E}_{(\tau,x)\sim P_F}\left[-2\log\frac{R(x)P_B(\tau|x)}{P_F(\tau;\theta)}\nabla_\theta\log P_F(\tau;\theta)\right], \tag{22}$$

which is estimated by sampling $\tau\sim P_F$ (terminating in $x\in\mathcal{X}$) and computing the term inside the expectation.

The model $P_F$ could also be optimized with respect to an evidence lower bound (ELBO) objective, i.e., minimizing $D_{\mathrm{KL}}(P_F(\tau)\|R(x)P_B(\tau|x))$. We derive the gradient of this KL:

$$\nabla_\theta D_{\mathrm{KL}}(P_F(\tau;\theta)\|R(x)P_B(\tau|x)) \tag{23}$$

$$= \nabla_\theta\mathbb{E}_{(\tau,x)\sim P_F}\left[\log\frac{P_F(\tau)}{R(x)P_B(\tau|x)}\right]$$

$$= \mathbb{E}_{(\tau,x)\sim P_F}\left[\nabla_\theta\log\frac{P_F(\tau;\theta)}{R(x)P_B(\tau|x)} + \log\frac{P_F(\tau;\theta)}{R(x)P_B(\tau|x)}\nabla_\theta\log P_F(\tau;\theta)\right]. \tag{24}$$

The last step is the standard score function trick, and the Reinforce estimator optimizes the KL by sampling $\tau \sim P_F(\tau; \theta)$ and using the term inside the expectation as the direction of the gradient step. But now notice that

$$\mathbb{E}_{(\tau,x) \sim P_F} \left[ \nabla_\theta \log \frac{P_F(\tau; \theta)}{R(x) P_B(\tau|x)} \right] = \mathbb{E}_{(\tau,x) \sim P_F} \left[ \nabla_\theta \log P_F(\tau; \theta) \right] = 0, \tag{25}$$

because of the constraint $\sum_\tau P_F(\tau; \theta) = 1$. We conclude that the expected trajectory balance gradient (22) is equal to the expected Reinforce gradient (24) up to a constant.

However in the vicinity of the optimum (when TB and KL equal 0), the TB graident estimator has lower variance, as the following computation shows:

$$\text{Var}_{(\tau,x) \sim P_F} \left[ \log \frac{p(x)}{P_F(\tau; \theta)} \nabla_\theta \log P_F(\tau; \theta) \right]$$

$$- \text{Var}_{(\tau,x) \sim P_F} \left[ \nabla_\theta \log \frac{P_F(\tau; \theta)}{R(x) P_B(\tau|x)} + \log \frac{P_F(\tau; \theta)}{R(x) P_B(\tau|x)} \nabla_\theta \log P_F(\tau; \theta) \right]$$

$$= -\mathbb{E}_{(\tau,x) \sim P_F} \left[ (\nabla_\theta \log P_F(\tau; \theta) \nabla_\theta \log P_F(\tau; \theta)^\top) \left( 1 + 2 \log \frac{R(x) P_B(\tau|x)}{P_F(\tau; \theta)} \right) \right].$$

If the term in parentheses is always positive (in particular, in the neighbourhood of the solution where $P_F(\tau) = R(x) P_B(\tau|x)$ for all $\tau$), then the difference of variances for all directional derivatives is negative for all $\theta$.

The connection between GFlowNets and variational methods was more thoroughly explored in two papers that appeared while this work was under review [31, 19].

# B Experimental appendix

## B.1 Hypergrid

For the GFlowNet policy model, we use an MLP of the same architecture as [3], with 2 hidden layers of 256 hidden units each. We train all models with a learning rate of 0.001 ($P_F$ and $P_B$ policy model) and 0.1 ($Z_\theta$) with up to $10^6$ sampled trajectories with a batch size of 16, using the Adam optimizer with all other parameters at their default values.

To reproduce the flow matching and non-GFlowNet baselines, we used the code published by [3] out of the box. For TB and DB, we used a learning rate of $10^{-3}$ for the flow and policy models and a $10^{-1}$ for the initial state flow $\log Z = \log F(s_0)$. (In a search of learning rates over powers of 10, $10^{-3}$ was found to be the largest that does not lead to instability in the form of rapid mode collapse.) All experiments with TB and DB were performed on CPU and take about 2 hours for $10^6$ episodes on a single core, totaling $\sim$10 CPU days for all 24 DB and TB experiment settings with 5 seeds each:

$$\{\text{TB}, \text{DB}\} \times \{\text{uniform } P_B, \text{learned } P_B\} \times \{R_0 = 10^{-1}, 10^{-2}, 10^{-3}\} \times \{(H, d) = (8, 4), (64, 2)\}.$$

## B.2 Molecule synthesis

We use the dataset and proxy model provided by [3]. We also train GFlowNet using the same architecture and hyperparameters (except $\beta$ and learning rate) but using the trajectory balance loss presented in this paper, using fixed uniform backward policy $P_B$. The binding scores in the provided dataset were computed with AutoDock [26].

To test hyperparameter robustness we trained models using reward exponents $\beta = \{4, 8, 10, 16\}$, and learning rates $\{5 \times 10^{-5}, 10^{-4}, 5 \times 10^{-4}, 10^{-3}\}$. In contrast to [3], we used a more exploratory training policy: with probability 0.1 (instead of the original 0.05) trajectories are set to stop at some length $k$, which is chosen uniformly between 3 and 8, the minimum and maximum allowed trajectory length respectively.

We observed a runtime improvement of up to $5\times$ for TB relative to FM. There are three factors responsible for this:

(1) Most importantly, FM requires as many model evaluations as there are parents of all states in a sampled trajectory, since the model gives the out-flows $F(s \rightarrow s')$ for an input state $s$, while the

objective involves a sum over in-flows. On the other hand, TB and DB require just one evaluation of the forward and backward policy models per state.

(2) The average trajectory length. If a model learns to terminate early with higher frequency, trajectories are shorter and fewer model evaluations are required.

(3) Hardware and the ratio of CPU and GPU load. Experiments on the molecule domain were run on a Tesla K80 GPU; the computation time benefit of TB appears to be smaller but still present on newer hardware with identical batch size settings. (Meanwhile, experiments on the lightweight grid domain were run on CPU, and the trajectory length was the main factor controlling runtime.)

## B.3   Bit sequence generation

**Generating reference sequences**. Let $H$ be a set of symbols (short bit sequences of length $b$), for instance $H = \{0110, 1100, 1111, 0000, 0011\}$. Sequences in $S$ are then constructed by randomly combining $m$ symbols from $H$, for instance, 0011110000000011 where $m = 4$. This construction imposes a structure on $R(x)$. In our experiments we set $m = \frac{n}{b}$, $b = 8$, $H = \{'00000000','11111111','11110000','00001111','00111100'\}$.

**Generating the test set**. Since the reward is defined based on the edit distance from the sequences in set $M$, we generate a test set sampled approximately uniformly over the possible values of $R(x)$ as follows: (1) pick a mode $s \in M$, (2) modify $i$ bits randomly $\forall i < n$ and we repeat this for all the modes.

**Implementation**. We implement GFlowNets with TB and FM in PyTorch for autoregressive generation tasks, along with the A2C baseline. For the MARS (MCMC) baseline we modify the implementation released by [3].

**Hyperparameters**. We use a Transformer [28] as the neural network architecture for all the methods. We use 3 hidden layers with hidden dimension 64 with 8 attention heads. All methods were trained for $50,000$ iterations, with a minibatch size of 16. We set the the random action probability to 0.0005 selected from $\{0.0001, 0.0005, 0.001, 0.01\}$, the reward exponent $\beta$ to 3 selected from $\{2, 3, 4\}$, and the sampling temperature for $P_F$ to 1 for the GFlowNets. For trajectory balance we use a learning rate of $1 \times 10^{-4}$ selected from $\{10^{-5}, 10^{-4}, 5 \times 10^{-4}, 10^{-3}, 5 \times 10^{-3}\}$ for the policy parameters and $1 \times 10^{-3}$ for $\log Z$. For flow matching we use a learning rate of $5 \times 10^{-4}$ selected from $\{10^{-5}, 10^{-4}, 5 \times 10^{-4}, 10^{-3}, 5 \times 10^{-3}\}$ with leaf loss coefficient $\lambda_T = 10$. For A2C with entropy regularization we share parameters between the actor and critic networks, and use learning rate of $10^{-4}$ selected from $\{10^{-5}, 10^{-4}, 5 \times 10^{-4}, 10^{-3}, 5 \times 10^{-3}\}$ with entropy regularization coefficient $10^{-3}$ selected from $\{10^{-4}, 10^{-3}, 10^{-2}\}$. For SAC we use the formulation in [6] with a learning rate of $5 \times 10^{-4}$ selected from $\{10^{-5}, 10^{-4}, 5 \times 10^{-4}, 10^{-3}, 5 \times 10^{-3}\}$ target network update frequency 500 and 200 initial random steps. For the MARS baseline we set the learning rate to $5 \times 10^{-4}$ selected from $\{10^{-5}, 10^{-4}, 5 \times 10^{-4}, 10^{-3}, 5 \times 10^{-3}\}$. For all the methods we use the Adam optimizer. Overall, for the Bit sequence generation experiments we used 25 GPU days.

## B.4   AMP generation

**Vocabulary**. The vocabulary of the 20 amino acids is defined as: ['A', 'C', 'D', 'E', 'F', 'G', 'H', 'I', 'K', 'L', 'M', 'N', 'P', 'Q', 'R', 'S', 'T', 'V', 'W', 'Y']

**Reward Model**. We use a Transformer-based classifier, with 4 hidden layers, hidden dimension 64, and 8 attention heads. We train it with a minibatch of size 256, with learning rate $10^{-4}$, with early stopping on the validation set.

**Hyperparameters**. As with the bit sequences, we use a Transformer [28] as the neural network architecture for all the methods. We use 3 hidden layers with hidden dimension 64 with 8 attention heads. All method were trained for $20,000$ iterations, with a mini batch size of 16. We set the the random action probability to 0.01 selected from $\{0.0001, 0.0005, 0.001, 0.01\}$, the reward exponent $\beta : R(x)^\beta$ to 3 selected from $\{2, 3, 4\}$, and the sampling temperature for $P_F$ to 1 for the GFlowNets. For trajectory balance we use a learning rate of $5 \times 10^{-3}$ selected from $\{10^{-5}, 10^{-4}, 5 \times 10^{-4}, 10^{-3}, 5 \times 10^{-3}\}$ for the flow parameters and $1 \times 10^{-2}$ for $\log Z$. For flow matching we use a learning rate of $5 \times 10^{-4}$ selected from $\{10^{-5}, 10^{-4}, 5 \times 10^{-4}, 10^{-3}, 5 \times 10^{-3}\}$ with leaf loss coefficient $\lambda_T = 25$. For A2C with entropy regularization we share parameters between the actor and critic networks, and use learning rate of $5 \times 10^{-4}$ selected from $\{10^{-5}, 10^{-4}, 5 \times 10^{-4}, 10^{-3}, 5 \times 10^{-3}\}$ with entropy

regularization coefficient $10^{-2}$ selected from $\{10^{-4}, 10^{-3}, 10^{-2}\}$. For SAC we use the formulation in [6] with a learning rate of $5 \times 10^{-4}$ selected from $\{10^{-5}, 10^{-4}, 5 \times 10^{-4}, 10^{-3}, 5 \times 10^{-3}\}$ target network update frequency $400$ and $200$ initial random steps. For the MARS baseline we set the learning rate to $5 \times 10^{-4}$ selected from $\{10^{-5}, 10^{-4}, 5 \times 10^{-4}, 10^{-3}, 5 \times 10^{-3}\}$. We run the experiments on 3 seeds and report the mean and standard error over the three runs in Table 1. Overall, for the AMP Generation experiments we used $14$ GPU days.