# OpenReview forum: "Trajectory balance: Improved credit assignment in GFlowNets"
_NeurIPS.cc/2022/Conference — NeurIPS 2022 Accept_

### Official Review · Reviewer_fSaD · 2022-07-01

**Rating:** 7
**Confidence:** 1
**Soundness:** 4 excellent
**Presentation:** 3 good
**Contribution:** 3 good

**Summary:**


Trajectory balance: Improved credit assignment in GFlowNets
In this paper, the authors improve the original generative flow networks’ objective function by modeling it on top of the entire trajectories generated from the policy network. This new objective function improves the convergence, diversity, robustness etc. compared to existing methods.


**Questions:**


1) I am entirely sure why the proposed algorithm improves “credit assignment”. Maybe the authors will elaborate on this? I do see improved results in terms of the correlations between final reward and policy probability across different tested environments & tasks.
But why does it improve the credit assignment? Maybe it improved something else instead?
Is there a metric or discussion to support this claim?

2) Does the algorithm scale to high-dimensional problems? A lot of the robotics tasks also involve actions that incrementally modify the state of the robot to solve the tasks (in gym for example). How well does the proposed algorithms work with these problems?
Also does the algorithm work with problems that have continuous state space?

3) The fonts of the figures are way too small (for example Figure 2, Figure 4), and it is hard to read without going back to the pdf version.



**Limitations:**

1) As far as I know, I don’t think it will have any potential negative societal impact directly.
2) In terms of the limitations of the paper itself, I think it would be great for the authors to comment on how well the algorithm will scale for high-dimensional robotics problems. As I have no related knowledge, I am not entirely sure how difficult the tasks of drug molecule synthesis and AMP generation are and how well the baselines should perform.



**Strengths And Weaknesses:**


Unfortunately I don’t have any experience or knowledge working with generative flow networks, and therefore I can only provide my evaluation and feedback as an outsider.

1) Originality: The paper discusses a relatively new technique of generative flow network and proposes a novel formulation of objective functions.

2) Quality & Clarity: The paper is very well written. The preliminary section provides complete and valuable background knowledge for readers not familiar with the topic.
The paper is supported both with solid math and well designed experiments. The results of the experiments are also fully discussed.

The code is given, and the results can be easily reproduced.
The comments and remarks are interesting and inspiring. The related work is very complete and adequate, and the connections to some other algorithms or methods are discussed.

3) Significance: In this paper, the generative flow networks are extended and potentially can be used in more complicated real-life problems. Although it is not clear how well the method can be scaled to high-dimensional continuous state problems such as robot arm target reaching tasks.

---

> ### Author Response · Authors · 2022-07-29
> **Author response**
>
> Thank you for your comments. We address the weaknesses, questions, and limitations below.
>
> > In this paper, the generative flow networks are extended and potentially can be used in more complicated real-life problems. Although it is not clear how well the method can be scaled to high-dimensional continuous state problems such as robot arm target reaching tasks. [...] Does the algorithm scale to high-dimensional problems? A lot of the robotics tasks also involve actions that incrementally modify the state of the robot to solve the tasks (in gym for example). How well does the proposed algorithms work with these problems? Also does the algorithm work with problems that have continuous state space?
>
> We agree that both in this paper and other work on GFlowNets [8,14,29], the problems studied have been less concrete or "real-life" than robot motion planning tasks. Although the authors are outsiders to robotics, we imagine that one of the main challenges would be in generalizing GFlowNets to continuous space and continuous time -- directions that we are very interested in and already studying.
> - Some theory for continuous-space GFlowNets is given in [4], where it is shown that the DB objective can be used for continuous state spaces. The TB objective is also applicable to continuous spaces, with the transition probabilities replaced by probability densities in equation (14). (We can add this to the paper.)
> - GFlowNets generating points in high-dimensional (up to 784-dim) *discrete* spaces were successfully trained in [29] using the TB objective. We foresee that very high-dimensional *continuous* spaces could require different action space designs, with each action updating several dimensions at once -- in fact, there is a connection between GFlowNets and fitting stochastic differential equations / stochastic optimal control. We are actively working on this direction, but it is outside the scope of this paper.
>
> > I am entirely sure why the proposed algorithm improves “credit assignment”. Maybe the authors will elaborate on this? I do see improved results in terms of the correlations between final reward and policy probability across different tested environments & tasks. But why does it improve the credit assignment? Maybe it improved something else instead? Is there a metric or discussion to support this claim?
>
> Please see the answer to Reviewer RAf4.
>
> > The fonts of the figures are way too small (for example Figure 2, Figure 4), and it is hard to read without going back to the pdf version.
>
> Thank you for pointing this out; we will reconsider the design of these figures to make them more readable for the camera-ready version.

---

### Official Review · Reviewer_Bg1m · 2022-07-11

**Rating:** 7
**Confidence:** 4
**Soundness:** 3 good
**Presentation:** 3 good
**Contribution:** 3 good

**Summary:**

This paper proposes a new learning objective for GFlowNets, trajectory balance TB), as a more efficient alternative to previously used objectives. A GFlowNet is a trained stochastic policy or generative model, trained such that it samples objects $x$ through a sequence of constructive steps, with probability proportional to $R(x)$, where $R$ is some given non-negative integrable reward function.

More specifically, GFlowNets aim to approximate a Markovian flow F on G such that

$F(x) = R(x)~~ \forall{x} \in X$

Generally speaking, a GFlowNet could be any learning algorithm consisting of:
(1) a model capable of providing the initial state flow and the forward action
distributions for any nonterminal state s, determining a Markovian flow F.
(2) an objective function, such that if the model is capable of expressing any action distribution and the
objective function is globally minimized, then the constraint is satisfied for the corresponding Markovian flow F .

This work focuses on finding a more appropriate objective function, distinct from those already tested and used in previous works. Namely,
Flow matching (FM) objective and Detailed balance (DB) objective. Both are inefficient in credit propagation across
long action sequences.

The proposed alternative is called "trajectory balance", as the title indicates. Basically, this approach relies on the "trajectory balance constraint", which in fact balances the Markovian flow F (and its forward and backward policies) over complete trajectories.

A set of experiments to test the performance of TB is provided in Section 5. More specifically: (1) The use of an Hypergrid environment to understand better the behavior with respect to MCMC and PPO, and more interestingly when using GFlowNets with FM and DB. (2) Experiment on Small drug molecule synthesis where TB, FM and DB are compared. (3) Autoregressive sequence generation in (3.1) bit sequences' problem and (3.2) Anti-Microbial Peptide (AMP) generation.


**Questions:**

1- One evident question would be: why publish this idea now? I mean, it results at least surprising that it has been already used in papers already published. I see all of them belong to 2022, so it may be for distinct reasons, but I feel curious about the chronological order in publication.

2- Runtime gain (numerical speedup) is only explicitly commented (x5) in the drug molecule synthesis problem. I assume this is because it is more significant there. However, how does it affect the other problems? Is there any quantitative estimation of the general profit with respect to FM and DB? What aspects/factors this improvement will depend on?

**Strengths And Weaknesses:**

Positive aspects:

* Well presented and original idea (within GFlowNets' world)
* Good quality and clarity, even though in the experimentation part there is little space for further analysis
* Already successfully applied (as indicated in the last paragraph of Section 1)

Less positive aspect:

* It could be a subjective point, but the contribution is quite specific to be applied in a relatively new "tool/model" and solving a quite particular task on it.

---

> ### Author Response · Authors · 2022-07-29
> **Author response**
>
> Thank you for your comments. We answer your questions below.
>
> > One evident question would be: why publish this idea now? I mean, it results at least surprising that it has been already used in papers already published. I see all of them belong to 2022, so it may be for distinct reasons, but I feel curious about the chronological order in publication.
>
> The chronological order is due to reasons outside our control, but we hope the "surprise" is seen as a strength and not a weakness!
>
> Generative flow networks give a new view on control as inference, or, as we like to think of it, variational inference via reinforcement learning. They were indeed introduced very recently (NeurIPS 2021), but have already been successfully used in very diverse settings beyond the toy grid and molecule domains: Bayesian structure learning [8], biological sequence design [14], and various discrete probabilistic modeling problems [29]. In each of these applications, trajectory balance or its generalizations have proven essential.
>
> > Runtime gain (numerical speedup) is only explicitly commented (x5) in the drug molecule synthesis problem. I assume this is because it is more significant there. However, how does it affect the other problems? Is there any quantitative estimation of the general profit with respect to FM and DB? What aspects/factors this improvement will depend on?
>
> This is a good question, and we will include extended discussion of computation costs in the final version. There are three aspects:
> - The key factor responsible for the speedup is that FM requires as many model evaluations as there are *parents* of all states in a sampled trajectory, since the model gives the out-flows $\log F(s\rightarrow s')$ for an input $s$, while the objective involves a sum over in-flows. On the other hand, TB and DB require just one evaluation of the forward and backward policy models per state. The speedup is greatest in the molecule domain because states can have a large number of parents, while on the grid domain the number of parents is 4 (for most states) and on the sequence domains it is 1, so we do not expect a speedup. (We can do additional benchmarking experiments to confirm this.)
> - An additional factor in the computation time per batch is the average trajectory length. If a model learns to terminate early with higher frequency, trajectories are shorter and fewer model evaluations are required.
> - Hardware and the ratio of CPU and GPU load should also be considered. Experiments on the lightweight grid domain were run on CPU, and the trajectory length was the main factor controlling runtime. Experiments on the molecule domain were run on a Tesla K80 GPU; the computation time benefit of TB appears to be smaller but still present on newer hardware with identical batch size settings.

---

### Official Review · Reviewer_F9mg · 2022-07-13

**Rating:** 7
**Confidence:** 2
**Soundness:** 3 good
**Presentation:** 3 good
**Contribution:** 3 good

**Summary:**

The paper proposes a new objective for GFlowNets and shows improved performance in experiments.

**Questions:**

Can we add more explanations on how to interpret figure 3?

**Limitations:**

N.A.

**Strengths And Weaknesses:**

Strengths:
1. Clear presentation and introduction of the background of GFlowNets.
2. Theoretical backup on the proposed objective.
3. Clear performance improvements.

Weaknesses:
1. Figures 2 and 3 are a bit hard to read.

---

> ### Author Response · Authors · 2022-07-29
> **Author response**
>
> Thank you for the comments. Regarding the weaknesses you pointed out, we will reconsider the design of Figures 2 and 3 to make them more readable, and we would be happy to take into account any concrete suggestions from you during the discussion period.
>
> We realize the discussion in Section 5.2 is quite dense and will add more details when given an extra page for the final version. For now, we give a clearer explanation of Figure 3:
> - Setup: We trained GFlowNets with the FM, DB, and TB objectives to generate molecules by incremental addition of building blocks consisting of a few atoms each. The GFlowNets were optimized to sample from a reward distribution given by a fixed, pretrained model of binding energy to a target. This replicates the setup of [3].
> - For a trained GFlowNet, the marginal likelihood $p_\theta(x)$ of sampling a given molecule $x$ can be computed. An optimal model should have $p_\theta(x)$ proportional to the reward $R(x)$ for all $x$, or, equivalently, $\log p_\theta(x)$ should equal $\log R(x)$ up to a constant independent of $x$. One way to measure the quality of fit is using the Pearson correlation of $\log p_\theta(x)$ and $\log R(x)$ computed on a held-out set of molecules $x$. This should equal 1 for a perfect model.
>   - The left figure plots $\log R(x)$ against $\log p_\theta(x)$ for $x$ in such a held-out set and shows that the Pearson correlation is highest for models trained with TB than with FM or DB, under optimal hyperparameter settings for each.
>   - The centre figure shows robustness of this correlation to hyperparameter choices: we trained models with each objective with multiple random seeds and range of hyperparameters and found that both the mean and maximum values of the correlation are highest for the TB objective.
> - The right figure illustrates a possible downstream benefit of TB training. In drug design, one often seeks *diverse* high-reward candidates, since a generated molecule may be impossible to synthesize, not actually effective against the target, have harmful effects, etc. Thus we measure the diversity of the top 1000 molecules generated by models trained with each objective. The lower score of TB indicates greater diversity. In comparison, FM models more often collapse to sampling a single high-reward molecule with high probability.

---

### Official Review · Reviewer_9Af4 · 2022-07-20

**Rating:** 8
**Confidence:** 3
**Soundness:** 4 excellent
**Presentation:** 3 good
**Contribution:** 3 good

**Summary:**

This paper proposes a new objective function for Generative Flow Networks (GFlowNet), which aims to generate objects with probability proportional to a given energy function. The proposed objective function is termed trajectory balance. This objective comes from a constraint that the flow of each complete trajectory should be consistent by either forward computation or backward compution. This constraint is formulated by Eq. (13), and is further converted to an objective function given by Eq. (14).

Trajectory balance suggests parameterizing the forward and backward probabilities (P_F and P_B) between connected states, as well as the flow volume Z of each state. The parameters can be trained by stochastic gradient descent. The authors further prove that, if Eq. (14) is minimized to zero, the GFlowNet can generate samples exactly from the target distribution. Experiments further show that trajectory balance outperforms the conventional two objective functions (flow matching and detailed balance) in terms of convergence, diversity of generated samples, and robustness to long action sequences and large action spaces.

**Questions:**

Please refer to the weakness.

**Limitations:**

Please refer to the weakness.

**Strengths And Weaknesses:**

**Strengths**:

This paper proposes an effective new objective function of training GFlowNets termed trajectory balance (TB). As a new learning method for generating complicated compositional objects, GFlowNet has recently gained an increasing research interest. Compared with conventional objectives, i.e., flow matching (FM) and detailed balance (DB), trajectory balance is empirically observed to overcome the difficulty of slow credit assignment, and gains a faster convergence speed and more diversified generated samples. This paper clearly sheds light on the research of generated flow methods, providing a new inspiring methodology for applying GFlowNet to various domains.

**Weakness**:

Although this paper demonstrates the success of TB through substantial experiments, it is not that clear why it performs better than FM and DB. It is argued that one benefit of TB is that the global minimizer can define a policy that samples exactly from the target distribution. However, this also holds for FM and DB. Another benefit claimed for TB is the improved credit assignment, but very few discussions are presented. Specifically, why does considering complete trajectories lead to better credit propagation in GFlowNet training?

Another concern is that the paper claims that the global minimizer of TB gives rise to exactly the target distributions. However, this is on the assumption that the parameterization of GFlowNet has sufficient representation power, i.e., Eq. (14) can be trained to exactly zero (Proposition 1). This may not always be the case if the task is too complicated or the training data is non-realizable. The authors may want to remind the readers of the above issue in the introduction and relevant discussions.

---

> ### Author Response · Authors · 2022-07-29
> **Author response**
>
> Thank you for your comments. We answer your questions below.
>
> > Although this paper demonstrates the success of TB through substantial experiments, it is not that clear why it performs better than FM and DB. It is argued that one benefit of TB is that the global minimizer can define a policy that samples exactly from the target distribution. However, this also holds for FM and DB. Another benefit claimed for TB is the improved credit assignment, but very few discussions are presented. Specifically, why does considering complete trajectories lead to better credit propagation in GFlowNet training?
>
> This is a good question. Although the benefits for TB are demonstrated *empirically* in the paper, we give two more points of view on this question, which we can emphasize more in the text:
> - An intuitive motivation for trajectory-level losses is that they, in contrast to local losses like FM and DB, propagate a reward signal along an entire sampled trajectory. In training with the FM and DB objectives, if a gradient update introduces a flow inconsistency at some state far from the root (such as when a novel high-reward state is sampled), the parent of this state must be visited again before the update is propagated any closer to the root, akin to the slow propagation of reward signals in temporal difference learning. This is not true of TB.
> - Another view comes from the derivation in Appendix A.3. There we show that the expectation of the on-policy TB gradient equals the gradient of a KL divergence between "forward" and "backward" trajectory distributions, a natural metric for tracking convergence of a learned distribution to a target. Thus TB gives an *unbiased* gradient of this divergence, while the FM and DB gradients are biased (although, as you noted, they also vanish at the global optimum).
>   - We suspect that a possible limitation of TB is the high *variance* of the gradient (relative to FM and DB objectives) for long action sequences and steep reward landscapes. Although the environments studied in this paper may be too simple for this limitation to surface, it can motivate research on interpolations between local and trajectory-level objectives, such as those discussed in Appendix A.2.
>
> > Another concern is that the paper claims that the global minimizer of TB gives rise to exactly the target distributions. However, this is on the assumption that the parameterization of GFlowNet has sufficient representation power, i.e., Eq. (14) can be trained to exactly zero (Proposition 1). This may not always be the case if the task is too complicated or the training data is non-realizable. The authors may want to remind the readers of the above issue in the introduction and relevant discussions.
>
> Yes, thank you for pointing this out. Like any neural policy algorithm, GFlowNets find an *approximation* to an optimal policy, which may not be exact if the model does not have sufficient representation power. Note that if the policies are represented in a tabular manner, i.e., the transition distributions from each state are learned as independent parameters, TB is, in fact, convex in the transition distributions' logits. On the other hand, a parametric estimate of the policy allows updates to one state to affect similar states, which can accelerate convergence and enable generalization to unseen states.

---

> > ### Comment · Reviewer_9Af4 · 2022-08-09
> > **Re: Rebuttal**
> >
> > The authors' responses fairly address my questions. I'm happy to maintain my score.

---

### Author Response · Authors · 2022-08-08
**Questions after author response?**

Dear reviewers,

Thank you again for all of your thoughtful comments. The end of the discussion phase is approaching, so we would like to ask if you have any additional questions for us after reading the responses.

Thank you,

The Authors

---

### Meta-Review · Area_Chair_3kBM · 2022-08-29

**Recommendation:** Accept
**Confidence:** Certain

**Metareview:**

The paper presents a new method for credit assignment in training GFlowNets, based on trajectory balance equations. All reviewers were in agreement the method offered theoretical guarantees as well as strong empirical results, and offers promise in terms of developing the growing field of Generative Flow Networks.

**Award:**

No

---

### Decision · Program_Chairs · 2022-09-14

Accept